# Optimizing Wildfire Prevention through the Integration of Prescribed Burning into 'Fire-Smart' Land-Use Policies

**Silvana Pais** [1,2,3,*], **Núria Aquilué** [4], **João P. Honrado** [1,2,3], **Paulo M. Fernandes** [5] and **Adrián Regos** [4]

1  Centro de Investigação em Biodiversidade e Recursos Genéticos, InBIO Laboratório Associado, Campus de Vairão, Universidade do Porto, 4485-661 Vairão, Portugal; jhonrado@fc.up.pt
2  Departamento de Biologia, Faculdade de Ciências, Universidade do Porto, 4099-002 Porto, Portugal
3  BIOPOLIS Program in Genomics, Biodiversity and Land Planning, CIBIO, Campus de Vairão, 4485-661 Vairão, Portugal
4  Centre de Ciència i Tecnologia Forestal de Catalunya (CTFC), Ctra. St. Llorenç de Morunys km 2, 25280 Solsona, Spain; adrian.regos@ctfc.cat (A.R.)
5  Centro de Investigação e de Tecnologias Agroambientais e Biológicas, CITAB, Inov4Agro, Universidade de Trás-os-Montes e Alto Douro, 5000-801 Vila Real, Portugal; pfern@utad.pt
*  Correspondence: silvana.pais@cibio.up.pt

**Abstract:** Integrating fire into land management is crucial in fire-prone regions. To evaluate the effectiveness and efficiency of prescribed fire (PF), we employed the REMAINS model in NW Iberia's Transboundary Biosphere Reserve Gerês-Xurés. We tested three levels of prescribed fire treatment effort for shrubland and grassland, employing three spatial allocation strategies: random distribution, prioritization in high-wildfire-risk zones, and creating fuel breaks by utilizing the existing road network. These approaches were assessed in isolation and in combination with three land-use scenarios: Business-as-usual (representing rural abandonment trends), High Nature Value farmland (reversing farmland abandonment), and Fire-Smart forest management (promoting fire-resistant landscapes). Our results confirm that PF is effective in reducing future wildfires (reductions up to 36%), with leverage values ranging from 0.07 to 0.45. Strategic spatial allocation, targeting wildfire-risk areas and existing road networks, is essential for maximizing prescribed fire's efficiency (leverage effort of 0.32 and 0.45; i.e., approximately 3 ha of PF decrease subsequent wildfire by 1 ha). However, the PF treatments yield the best efficiency when integrated into land-use policies promoting 'fire-smart' landscapes (reaching leverage values of up to 1.78 under policies promoting 'HNVf and 'fire-smart' forest conversion). These recommendations strengthen wildfire prevention and enhance landscape resilience in fire-prone regions.

**Keywords:** integrated fire management; fuel breaks; fuel management; Mediterranean-type ecosystems; fire-landscape dynamics; prescribed fire planning

## 1. Introduction

Fire is a key ecological process in many ecosystems that determines a wide range of ecosystem attributes [1]. From cycling nutrients to shaping vegetation structure, fire plays a key role in shaping landscapes and biodiversity patterns around the world [1–3]. However, the current increase in large wildfires is a serious ecological and socioeconomic threat [4–6]. In Southern Europe, the most-affected countries (Portugal, Spain, Italy, Greece, and France) base their fire management strategies on highly effective fire suppression to reduce total burnt areas [7–10]. Despite considerable efforts in fire suppression, there has been a persistent increase in both the size and severity of wildfires in Mediterranean Europe [7], fueled by factors like land abandonment and climate change [5,11,12]. The interplay between land abandonment and fire-exclusion policies, coupled with changing climatic conditions, has disrupted the fire regime in the mountain landscapes of Southern Europe [13–16]. In many regions, rural areas have been depopulated due to migration



to urban areas with the subsequent decline in traditional agroforestry activities [17–19]. Abandoned farmland and forests can become overgrown with vegetation that is more fire-prone, while the absence of traditional land-use practices, such as grazing, can also contribute to fuel buildup for extreme wildfires [16,19,20].

Some authors consider that the systematic extinction of all wildfires allows for an accumulation of fuel that will be burned in future larger and more extreme wildfires [21]. If large fires are a direct consequence of fuel buildup (among other factors), then one potentially efficient way to decrease their likelihood is through fuel reduction [22,23]. The need to increase prevention efforts is well recognized [24,25], but its effects should be durable over time and ecologically and economically sustainable. Prevention strategies may consist of networks of fuel breaks and low fuel-load mosaics strategically located to compartmentalize wildland areas and support fire suppression efforts [12,26,27]. However, mechanical fuel treatments are costly in terms of human and material resources, which limits their implementation. Fuel-reduction treatments should be therefore carefully planned prior to implementation to increase their cost–benefit ratio (e.g., [28–30]).

Prescribed fire (PF) can decrease surface and ladder fuels, which subsequently reduces potential fire behavior and the risk of crown fire and spotting [31], thus reinforcing fire prevention policies [32–34]. Fighting fire with fire represents an important paradigm shift after decades of fire-exclusion policies, with widely demonstrated positive effects in terms of fire risk reduction, especially in fire-adapted ecosystems [28,35–37]. Prescribed fire involves the planned and controlled use of fire, following a predefined prescription, to achieve specific and well-defined resource management goals. This approach aligns with state-of-the-art fire behavior and fire ecology knowledge [38]. Fire management planning aimed at reducing wildfire spread is a complex problem that requires careful consideration of landscape treatments in terms of spatial configuration and treatment density [37,39,40]. Prescribed fire is practiced in a wide diversity of vegetation types in Europe [32,34], South Africa [41], North America [42], and Australia [43,44]. Studies have shown that when implemented properly, it can help restore ecosystem health [45], reduce the buildup of flammable vegetation [40], enhance habitat conditions for certain species [45,46], and promote ecological diversity [47].

Prescribed fire effectiveness varies depending on vegetation and treatment types [28,48]. In open, single-layered vegetation types, prescribed fire can be stand-replacing, removing the grass or shrub layer while leaving unburnt patches. However, in forest areas with multiple fuel layers, PF is typically conducted as an understory low-intensity fire, consuming litter, downed woody fuels, grasses and shrubs on the forest floor, and understory. Additionally, depending on tree size and fireline intensity, PF may have a pruning effect, reducing the likelihood of subsequent crown fire development. This distinction highlights the short-lasting treatment effect in grassland or shrubland compared to forests, where treatment effectiveness persists as long as crowning is avoided [28].

In this work, we assess the effectiveness of PF as a management tool to reduce the fire hazard in a mountain area strongly affected by wildfires and land abandonment, the Transboundary Biosphere Reserve Gerês-Xurés (NW Iberia), considering the current fire-suppression levels and plausible future land-use scenarios. We used the REMAINS model that couples a fire–vegetation dynamic model that simulates wildfires and vegetation dynamics, with a land-use change model [49]. The most-updated version of the model incorporates a submodule to simulate PF, which allows us to reproduce areas treated with planned fires, thus creating wildfire suppression opportunities in strategically selected locations. In this study, we considered three PF strategies embedded into three land-use policy scenarios. In particular, we simulated three levels of treatment effort (affecting 0.5, 5, and 10% of the landscape) under three PF allocation strategies: (1) using the road network to create fuel breaks; (2) targeting areas with a higher fire hazard; and (3) treatments randomly distributed across the landscape. These PF strategies were simulated under three plausible land-use management scenarios: (1) Business-as-usual (BAU), which represents the most recent rural abandonment trend; (2) High Nature Value farmland (HNVf), which assumes

EU agricultural policies promoting extensive agropastoral activities; and (3) Fire-Smart (FS) forest conversion toward more fire-resistant stands (i.e., from fast-growing tree plantation to native oak forest). In this simulation exercise, PF is strictly focused on only two land-cover types: shrubland and grassland, due to their large coverage (35%) in the study area and the low social acceptance of PF programs in private forest lands. By studying the amount of land managed by PF, the different allocation strategies, and their outcomes, we aimed to understand how PF can be strategically incorporated to enhance overall land management and wildfire hazard reduction.

## 2. Materials and Methods

### 2.1. Study Area

The study area was the Transboundary Biosphere Reserve Gerês-Xurés (BR-GX; c.a. 276,000 ha, of which 71% is in Portugal and the remaining 29% is in Spain) in the NW Iberian Peninsula (Figure 1). The reserve encompasses four Natura 2000 sites and two national protected areas (Peneda-Gerês National Park in Portugal and Baixa Limia-Serra do Xurés Natural Park in Spain) [12]. This mountain region is topographically complex and diversified (Figure 1). The climate is predominantly temperate oceanic, characterized by high precipitation levels (mostly concentrated in autumn and winter), although the heterogeneous terrain and wide-ranging elevation allow diverse microclimatic conditions.

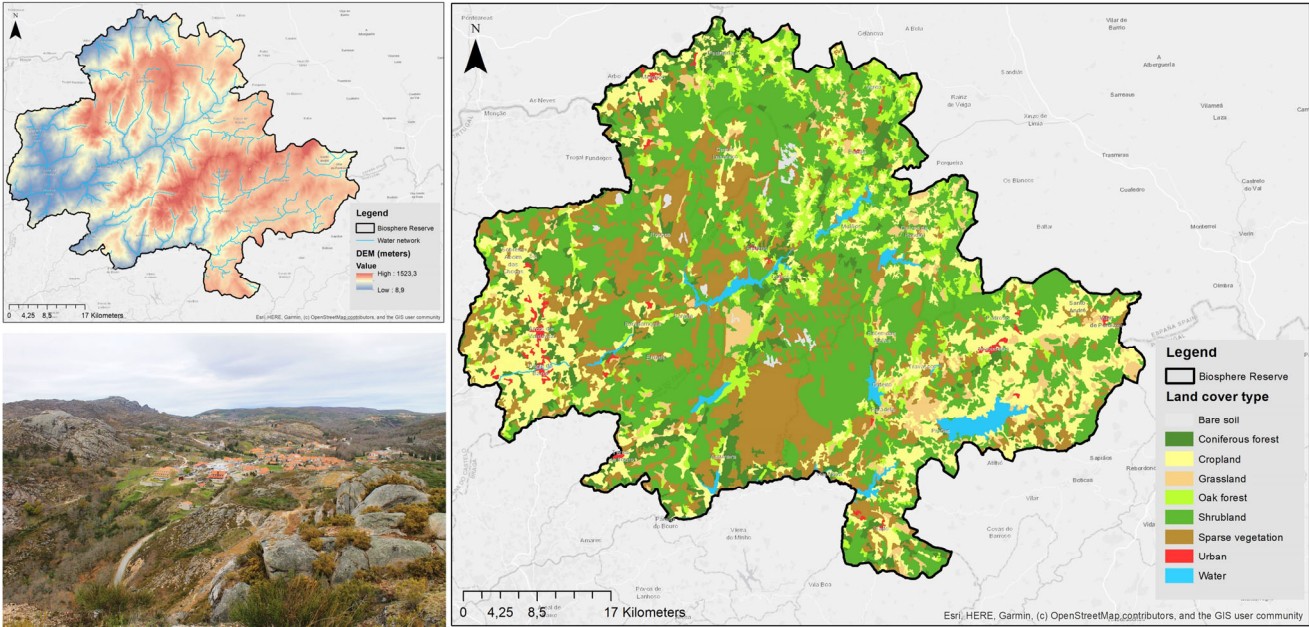

**Figure 1.** Altitude gradient and land-use and land-cover types of the study area (Photo by: Silvana Pais).

Currently, the landscape is dominated by shrubland, covering approximately 31% of the area, which accounts for more than 86,000 hectares. Additionally, there are fragmented deciduous forests spanning about 10.8%, equivalent to 29,800 hectares, mostly represented by *Quercus robur* and *Q. pyrenaica*. Furthermore, coniferous forests extend over approximately 17%, encompassing around 47,000 hectares, dominated by *Pinus sylvestris* and *P. pinaster* (Figure 1). Sparse vegetation (including rocky areas) is a highly relevant land-cover category in the study area, representing about 24%, i.e., 65,000 hectares. Like in other mountain areas in Southern Europe, population has decreased due to the abandonment of traditional agricultural and livestock activities and the aging of the remaining farmers [12]. Despite these changes, we can still find some agricultural areas (8%, i.e., more than 20,800 hectares) essentially located near the urban areas, as well as the presence of grasslands, although in a reduced percentage (4%) (Figure 1).

Over the last 30 years, landscape changed primarily with the decline in pasture areas (approximately 36,000 hectares have been lost from 1990 to the present day, representing a loss of over 13%) and agricultural spaces (a reduction of nearly 3%, equivalent to 7500 hectares over 30 years), together with an increase of nearly 10%, 5%, and 2% in coniferous forests, deciduous forests, and urban areas, respectively.

Wildfires have occurred on a large scale and with high frequency in both countries (Figure 2). Over the past 30 years, RBGX has witnessed the burning of approximately 256,000 hectares, of which more than 199,000 hectares were in Portugal, and there have been years with exceptionally severe wildfires, where the burnt area exceeded 6, 7, and up to 8% of the land. On average, about 3.1% of the territory burns every year (Figure 2). The Spanish side, on the other hand, has an average annual burnt area of 2.3% relative to its territory. However, it has also experienced severe years, such as 1998, 2011, and 2017 (Figure 2).

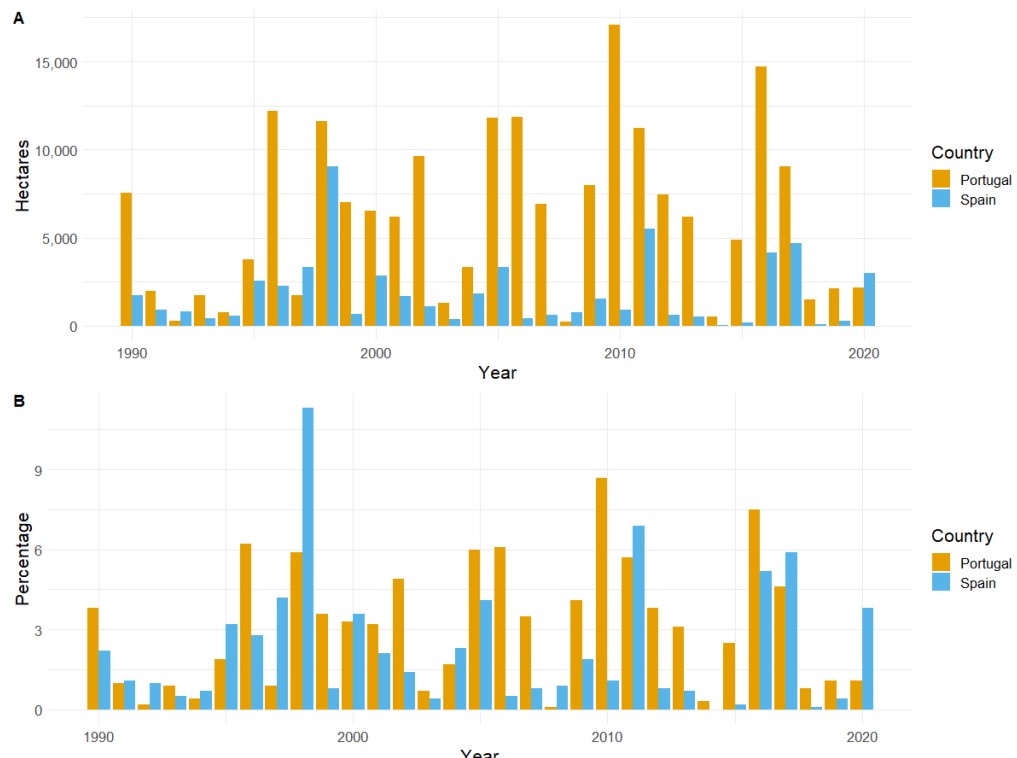

**Figure 2.** Wildfire extent in the study area over the last 30 years (1990–2020). Panel (**A**) represents the area burned in hectares and panel (**B**) the percentage of area burned in the study area in the northwestern Spanish province of Galicia and in Portugal [50,51].

Based on the historical wildfire data, burn probability is high in certain regions of both countries, with some critical locations having an approximate fire return interval of three years [52].

### 2.2. Modeling Framework and Management Scenarios

#### 2.2.1. Conceptual Framework

The use of fire as preventive tool has been clearly undermined by the current fire suppression paradigm in Southern Europe, which aims to exclude all types of fires from the landscape without considering their ecological and socioeconomic roles [36]. Despite these fire-exclusion policies, some regions are often subjected to frequent human-induced wildfires with various causes (e.g., vandalism, arson, revenge, land-use change attempts, territorial disputes, and mental problems) but rooted in profound socioeconomic changes [53–55]. Despite the growing concern over the conservation of natural resources of this area (espe-

cially within protected area), the current wildfire suppression strategies have been found to be insufficient to deal with large fires [56,57]. This ineffectiveness was associated with several factors: (1) low accessibility during firefighting operations, (2) topography of the area that favors the rapid spread of wildfires, especially in extremely adverse fire weather conditions, and (3) the current fire-suppression paradigm [5] due to legislative constraints (ecological restrictions, for example, that may prevent fuel-reduction treatments).

To quantify the potential impact of alternative fire management and land-use policies on the fire regime, we used a dynamic fire-landscape model. The fire-landscape simulations allowed quantifying the impact of fire and land management on fire regime (namely, burned and suppressed areas) under each management scenario, as well as the temporal dynamics of the main land-cover types.

### 2.2.2. The REMAINS Model

REMAINS is a spatially explicit model that integrates the main factors driving fire-landscape dynamics in mountain landscapes of Southern Europe [12]. It includes the main anthropogenic and natural (abiotic and biotic) drivers of landscape change to study their spatiotemporal interactions and feedback effects at short and medium timescales. REMAINS model (implemented in the R language: https://github.com/FirESmart-Project/REMAINS (accessed on 1 November 2023)) allows the simulation of land-cover changes (agriculture abandonment or intensification) and forest-type conversions; wildfires (including fire ignition, spread, and extinction) and fire suppression; prescribed fire; and vegetation dynamics (natural succession and postfire regeneration) [49].

The REMAINS model reproduces fire-landscape dynamics according to predesigned scenario storylines, currently initialized and calibrated for the Transboundary Biosphere Reserve Gerês-Xurés. The primary objective of this study was to assess the effectiveness of PF as a management tool in mitigating the impact of wildfires, while also exploring strategies for distributing it throughout the landscape. Simultaneously, we aimed to test scenarios that integrate PF with forest conversion and agricultural promotion, assessing the potential for these combined strategies to substantially reduce the extent of areas at risk of future wildfires.

The model incorporates the simulation of planned fires, which are strategically employed to reduce fuel load and manage fuel breaks, creating additional opportunities for fire suppression in specific locations. The user has the flexibility to determine two key aspects: (i) the extent of area to be treated annually with PF and (ii) the criteria for selecting treatment's locations. Model parameters allow defining a time window during which past wildfires or PF can be used as suppression opportunities, and the extent of burning can be adjusted based on the characteristics of the ecosystem [49].

### 2.2.3. Prescribed Fire Scenarios

We aimed at designing fuel treatments based on PF within broader, landscape-scale objectives. Thus, we designed PF strategies (as a combination of fuel treatment effort and spatial configuration) to accomplish three general objectives (Table 1): (i) generate agroforest mosaics in the landscape that would increase firefighting capacity to suppress them; (ii) promote and renew rangelands to encourage extensive pastoral activities and thus meet needs of rural mountain communities; and (iii) contribute to an integrated management toward 'fire-smart' landscapes by managing fuel age and composition.

**Table 1.** Spatial framework for landscape fuel management. The strategies simulated in this study combined 3 fundamental aspects: (i) the need for landscape-scale management; (ii) burning objectives; and (iii) treatment strategy. Black-shaded areas in the maps depict fuel treatment areas (via prescribed fire) under three spatial strategies (see illustrative maps): (1) creating fuel breaks scattered across the landscape (mosaics), (2) fuel management in medium–high fire hazard areas and, (3) fire containers taking advantage of road networks to promote larger fuel breaks (adapted from [58]).

| Spatial Strategies for Prescribed Fire | | |
| --- | --- | --- |
| **Landscape Mosaics** | **Hazard Management** | **Strategic Containment** |
| Scenario | PF_rnd | PF_hzd | PF_road |
| Landscape goal | Disrupt wildfire spread and facilitate containment | Local protection of assets | Contain large wildfires at defensible locations |
| Performance measure | Reduction in landscape burn probability | Reduction in high fuel load | Area burned by prescribed fire |
| Burning objective | Reduce fire-spread rate | Reduce fire hazard and facilitate suppression | Establishes locations for containment and facilitates suppression |
| Treatment strategy | Landscape-oriented approach | Fire hazard-oriented approach | Firefighting-oriented approach |
| Example map |  (1) |  (2) |  (3) |

PF effectiveness depends on treatment effort (here using % of area treated as proxy), the duration of its effect, spatial planning plus vegetation type, topography, and fire weather conditions. In this study, we tested three levels of treatment effort by burning 0.5, 5, and 10% of the landscape per year. We tested the currently practiced scenario of burning 0.5% of the landscape [35]; then, we tested the scenario of the operationally viable amount to be treated (5% of the landscape [28]) and finally a theoretical high-effort scenario (10% of the landscape [28]), which, given the current context of the territory, was not included in our combined scenario simulations. Simultaneously, we tested three PF allocation strategies: (1) the random use of fire across the landscape until a certain annually treated area is reached ('PF_rnd'); (2) burning shrubland and grassland where fuel load is high ('PF_hzd'), i.e., areas that did not burn for a relatively extended period neither by unplanned nor planned fire; and (3) taking advantage of the road network to establish fuel breaks ('PF_road').

In all the scenarios, we assumed that an area burned by PF cannot be treated again for at least the next 4 years [40,59,60]. We quantified the effect of PF on the reduction in annual wildfire extent and its leverage. Conceptually, leverage, as defined by [61], refers to the total area protected from wildfire per unit area treated by fuel-reduction measures. Specifically, in the case of PF, leverage is calculated by examining the relationship between the extent of wildfire and the implementation of PF [62,63]. It can be expressed as 'return-for-effort', meaning that a leverage of, for example, 0.25 (1:4) indicates that treating 4 hectares with prescribed fire will lead to a 1-hectare decrease in wildfire extent [35].

The PF strategies were tested under four land-use policy scenarios.

### 2.2.4. Land-Use Policy Scenarios

We simulated the twelve above-defined PF scenarios under certain landscape management conditions for the near future (2021–2050) (Figure 3): (1) *Business-as-usual*, which represents a future landscape derived from the historical fire regime and land-use change trends that occurred in the recent past (1987–2010) [12]; (2) *High Nature Value farmland*

(HNVf) that should envisage a recovery of upland grasslands and croplands [12] to predict the potential impacts of a successful implementation of the EU environmental and rural policies on fire regime and nature conservation [64,65] as a counterpoint to the current BAU scenario; (3) *Fire-smart*, aiming at a 'fire-smart' forest conversion from plantations of fast-growing tree species (mostly *Pinus* and *Eucalyptus* spp. in the study area) to native forest (i.e., oak woodlands) to foster more fire-resistant (less-flammable) and/or fire-resilient landscapes [57]; and (4) *High Nature Value farmland + Fire-smart* that envisages an integrated management policy that combines the promotion of more resistant and less-flammable landscapes with policies aimed at gradually increasing agricultural areas, as an opportunity for fire suppression and farmland/grassland biodiversity conservation [12].

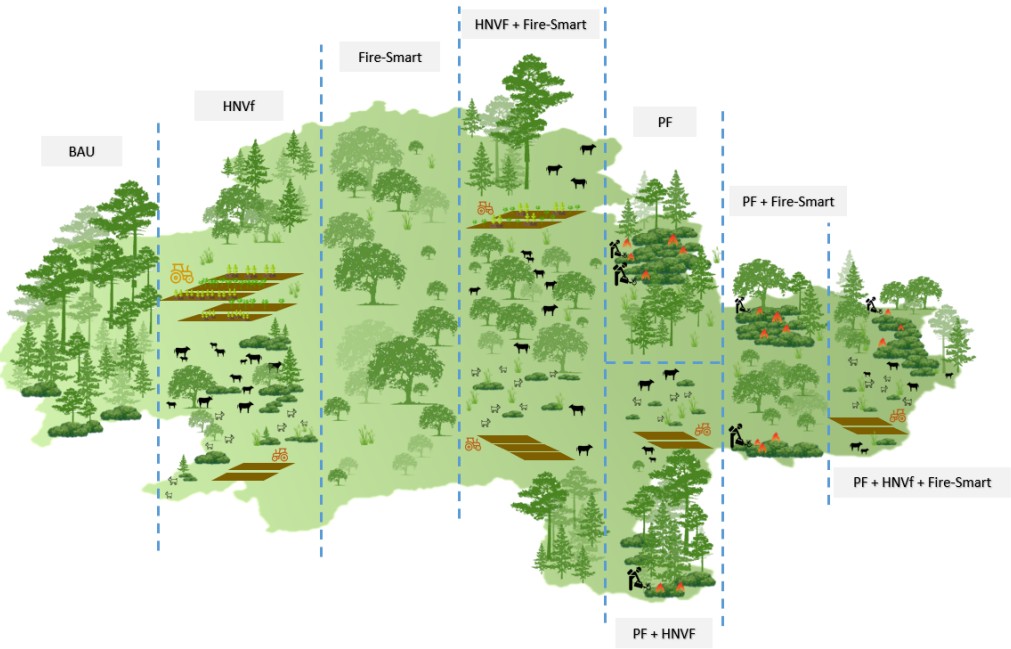

**Figure 3.** Representation of the main landscape management scenarios. Acronymous: BAU—*Business-as-usual*, HNVf—High Natural Value farmland, and PF—Prescribed Fire.

Scenario parameters regarding land-cover transitions were established considering historic land-use/-cover changes (1990–2020). The annual conversion rate from cropland to shrubland (rural abandonment rate) was set at 200 hectares and 180 hectares for grassland for the future simulations (Table 2). In the future 'HNVf' scenarios, the land abandonment rate was set to 0 and the annual conversion rate from shrubland to cropland and grassland was increased to 250 and 300 ha/year, respectively (see Table 2). In 'Fire-smart' scenarios, the conversion of coniferous into deciduous woodlands was also implemented to convert the total area of the coniferous forest area into deciduous (rate of 1, see [12]). To deal with the fire stochasticity, 30 replicates were performed for each scenario for a 30-year period (2021–2050).

**Table 2.** Management scenarios and annual land-cover conversion rates. 'RateOak' refers to the annual conversion rate from shrubland to oak (in %); 'RAb' is rural abandonment, for which the user defines how many annual hectares are converted through abandonment processes (in ha); 'PAb' means pasture abandonment, i.e., how many hectares per year are converted from grassland to sparse vegetation (in ha); 'Agriconv' means the conversion area from shrubland to cropland (in ha); 'PastureConv' means the conversion area from sparse vegetation to grassland (in ha); 'ActFireSmart' is the conversion of coniferous forest to deciduous woodlands (in %); 'PF' means Prescribed Fire (in ha).

| Scenario | RateOak | RAb | PAb | AgriConv | PastureConv | ActFireSmart | PF | Strategy |
|---|---|---|---|---|---|---|---|---|
| Rural abandonment | | | | | | | | |
| BAU | 1.6 | 200 | 180 | 0 | 0 | 0 | 0 | Null |
| PF0.5_rnd_BAU | 1.6 | 200 | 180 | 0 | 0 | 0 | 1000 | Random |
| PF5_rnd_BAU | 1.6 | 200 | 180 | 0 | 0 | 0 | 10,000 | Random |
| PF10_rnd_BAU | 1.6 | 200 | 180 | 0 | 0 | 0 | 20,000 | Random |
| PF0.5_hzd_BAU | 1.6 | 200 | 180 | 0 | 0 | 0 | 1000 | Fire hazard |
| PF5_hzd_BAU | 1.6 | 200 | 180 | 0 | 0 | 0 | 10,000 | Fire hazard |
| PF10_hzd_BAU | 1.6 | 200 | 180 | 0 | 0 | 0 | 20,000 | Fire hazard |
| PF0.5_road_BAU | 1.6 | 200 | 180 | 0 | 0 | 0 | 1000 | Road network |
| PF5_road_BAU | 1.6 | 200 | 180 | 0 | 0 | 0 | 10,000 | Road network |
| PF10_road_BAU | 1.6 | 200 | 180 | 0 | 0 | 0 | 20,000 | Road network |
| High Natural Value farmland | | | | | | | | |
| PF0.5_rnd_HNVf | 1.6 | 0 | 0 | 250 | 800 | 0 | 1000 | Random |
| PF5_rnd_HNVf | 1.6 | 0 | 0 | 250 | 800 | 0 | 10,000 | Random |
| PF0.5_hzd_HNVf | 1.6 | 0 | 0 | 250 | 800 | 0 | 1000 | Fire hazard |
| PF5_hzd_HNVf | 1.6 | 0 | 0 | 250 | 800 | 0 | 10,000 | Fire hazard |
| PF0.5_road_HNVf | 1.6 | 0 | 0 | 250 | 800 | 0 | 1000 | Road network |
| PF5_road_HNVf | 1.6 | 0 | 0 | 250 | 800 | 0 | 10,000 | Road network |
| Fire-Smart | | | | | | | | |
| PF0.5_rnd_FS | 2.4 | 200 | 180 | 0 | 0 | 1 | 1000 | Random |
| PF5_rnd_FS | 2.4 | 200 | 180 | 0 | 0 | 1 | 10,000 | Random |
| PF0.5_hzd_FS | 2.4 | 200 | 180 | 0 | 0 | 1 | 1000 | Fire hazard |
| PF5_hzd_FS | 2.4 | 200 | 180 | 0 | 0 | 1 | 10,000 | Fire hazard |
| PF0.5_road_FS | 2.4 | 200 | 180 | 0 | 0 | 1 | 1000 | Road network |
| PF5_road_FS | 2.4 | 200 | 180 | 0 | 0 | 1 | 10,000 | Road network |
| High Natural Value farmland + Fire-Smart | | | | | | | | |
| PF0.5_rnd_HNVf_FS | 2.4 | 0 | 0 | 250 | 800 | 1 | 1000 | Random |
| PF5_rnd_HNVf_FS | 2.4 | 0 | 0 | 250 | 800 | 1 | 10,000 | Random |
| PF0.5_hzd_HNVf_FS | 2.4 | 0 | 0 | 250 | 800 | 1 | 1000 | Fire hazard |
| PF5_hzd_HNVf_FS | 2.4 | 0 | 0 | 250 | 800 | 1 | 10,000 | Fire hazard |
| PF0.5_road_HNVf_FS | 2.4 | 0 | 0 | 250 | 800 | 1 | 1000 | Road network |
| PF5_road_HNVf_FS | 2.4 | 0 | 0 | 250 | 800 | 1 | 10,000 | Road network |

## 3. Results

### 3.1. Prescribed Fire Use in the Current Context of Rural Abandoment

Our simulations point to an increase in the extent of areas affected by wildfires in the coming years under the current abandonment conditions. We anticipate an annual increase of more than 1500 ha burned between 2021 and 2050 under the BAU scenario (Figure 4). In contrast, the implementation of PF holds promise in mitigating future wildfire impacts by reducing the extent of wildfire (Figure 4).

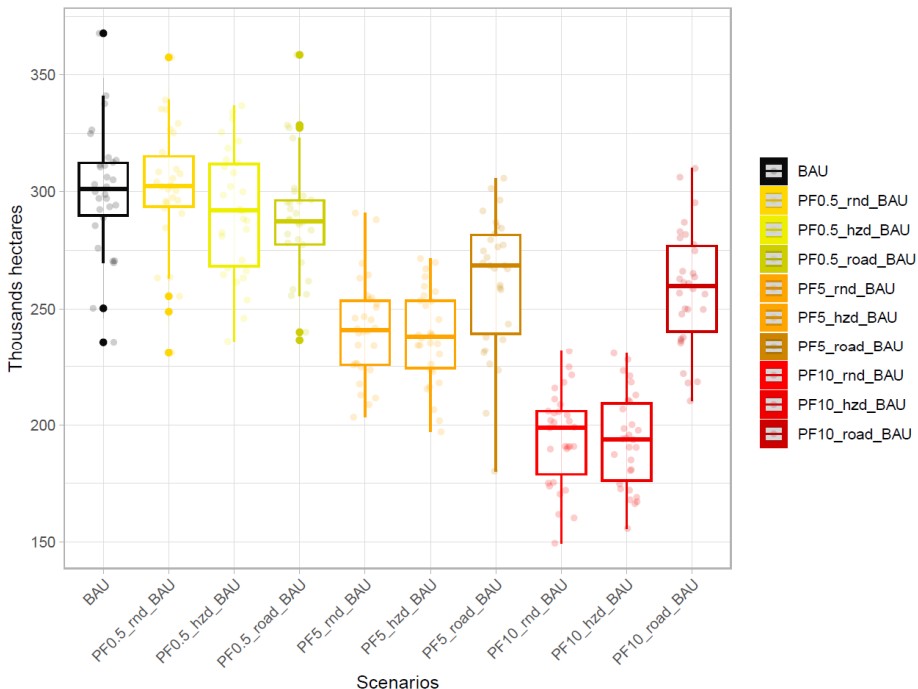

**Figure 4.** Predicted wildfire extent for 2021–2050 under different PF effort scenarios. The BAU scenario represents the annual trend of abandonment, wildfires, and fire-exclusion policies. The remaining scenarios represent the potential impact on burned area reduction through fuel management (shrubs and grasslands) under specific treatment efforts and PF allocation. See scenario acronyms in Table 2.

According to our simulations, current practices of low PF effort ('0.5' scenarios, i.e., burning 1000 hectares annually) reduce the wildfire extent by between 3 and 4.5%, corresponding to around 9600 and 13,600 ha under the 'risk' and 'road' strategies, respectively, over a 30-year period. Yet, no significant differences between fire allocation strategies were predicted under this scenario (see Figure 4). These scenarios exert a leverage effect of about 0.32 and 0.45, thereby implying that the wildfire extent can be decreased by 1 ha by treating 2–3 ha (Table 3).

When fuel treatment was carried out on 10,000 hectares annually (i.e., '5' scenarios), our simulations indicate decreases in the wildfire extent of about 60,300 to 63,200 hectares over 30 years (reductions of 20% to 21% in the 'PF5_rnd_BAU' and 'PF5_hzd_BAU' scenarios, respectively). The 'PF5_road_BAU' scenario would reduce significantly the less-burned area than previous ones (reductions of only 11%, corresponding to 32,600 hectares, see Figure 4). In summary, burning 10,000 hectares with PF would result in annual reductions in the wildfire extent of over 2000 hectares, i.e., a leverage of 0.2 (see scenarios 'PF5_rnd_BAU' and 'PF5_hzd_BAU', Figure 4 and Table 3), while the lowest-performing scenario shows a leverage of 0.1 ('PF5_road_BAU', Table 3).

**Table 3.** Leverage by management scenarios under rural abandonment conditions, comparing strategies, efforts, and treatment allocations.

| Spatial Strategy | Scenario | Area Treated (ha/Year) | Annual Reduction in Wildfire Area (ha) | % Annual Reduction (30-Year Period) | Leverage |
|---|---|---|---|---|---|
| Random | PF0.5_rnd_BAU | 1000 | - | - | - |
| Road | PF10_road_BAU | 20,000 | 1376.7 | 14 | 0.07 |
| Road | PF5_road_BAU | 10,000 | 1086.7 | 11 | 0.11 |
| Random | PF10_rnd_BAU | 20,000 | 3406.7 | 34 | 0.17 |
| Hazard | PF10_hzd_BAU | 20,000 | 3576.7 | 35.6 | 0.18 |
| Random | PF5_rnd_BAU | 10,000 | 2010 | 20 | 0.2 |
| Hazard | PF5_hzd_BAU | 10,000 | 2106.7 | 21 | 0.21 |
| Hazard | PF0.5_hzd_BAU | 1000 | 320 | 3 | 0.32 |
| Road | PF0.5_road_BAU | 1000 | 453.3 | 4.5 | 0.45 |

As expected, the scenarios with the highest PF effort (20,000 ha/year) showed significant reductions compared to the other scenarios. Overall, burning shrubland and pastures randomly can reduce the wildfire extent by approximately 34% (see 'PF10_rnd_BAU' scenario, Figure 4), with a leverage of 0.17 (Table 3). In scenarios targeting high fire-hazard areas, reductions in the wildfire extent can reach 36% (see 'PF10_hzd_BAU' scenario, Figure 4), with a leverage of 0.18 (Table 3). In the study area, annually burning 20,000 hectares with PF, taking advantage of the road network, has beneficial impacts on reducing the wildfire extent but with a smaller effect when compared to the performance of the 10,000-hectare scenarios (see '5' scenarios, Figure 4).

### 3.2. Prescribed Fire under Land-Use Policy Scenarios

The predicted burnt area was found to strongly rely on land-use management scenario (Figure 5). Our results indicate that combining prescribed burning and landscape management strategies can have a significant impact on the fire regime.

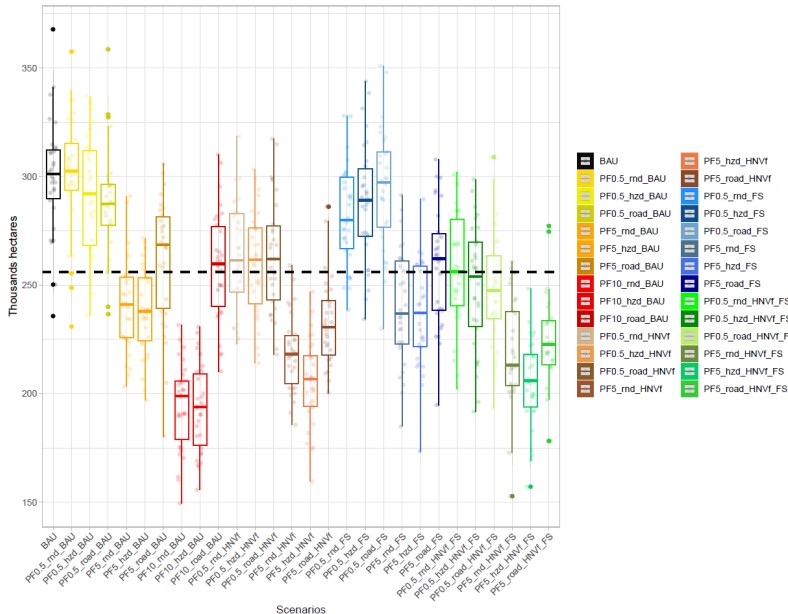

**Figure 5.** Predicted wildfire extent under each management scenario between 2021 and 2050. For all boxplots, lower and upper whiskers encompass the 95% interval, lower and upper hinges indicate the first and third quartiles, and the central line indicates the median value (solid dots are outliners). See scenario acronyms in Table 2.

In general, the scenarios showed reductions in wildfire areas ranging from 130 hectares annually in the lowest-performing option ('PF0.5_road_FS') to over 3100 hectares annually in the highest-performing option ('PF5_hzd_HNVf_FS', see Figure 5 and Table 4).

**Table 4.** Wildfire extent reduction and leverage by different fire management scenarios, comparing strategies, efforts, and treatment allocations.

| Spatial Strategy | Scenario | Area Treated (ha/Year) | Annual Reduction in Wildfire Area (ha) | % Annual Reduction 30-Year Period | Leverage |
|---|---|---|---|---|---|
| Road | PF0.5_road_FS | 1000 | 133.3 | 1 | 0.13 |
| Road | PF5_road_FS | 10,000 | 1300 | 13 | 0.13 |
| Random | PF5_rnd_FS | 10,000 | 2140 | 21 | 0.21 |
| Hazard | PF5_hzd_FS | 10,000 | 2140 | 21 | 0.21 |
| Road | PF5_road_HNVf | 10,000 | 2343.3 | 23 | 0.23 |
| Road | PF5_road_HNVf_FS | 10,000 | 2616.7 | 26 | 0.26 |
| Random | PF5_rnd_HNVf | 10,000 | 2773.3 | 28 | 0.28 |
| Random | PF5_rnd_HNVf_FS | 10,000 | 2943.3 | 29 | 0.29 |
| Hazard | PF5_hzd_HNVf | 10,000 | 3146.7 | 31 | 0.31 |
| Hazard | PF5_hzd_HNVf_FS | 10,000 | 3173.3 | 32 | 0.32 |
| Hazard | PF0.5_hzd_FS | 1000 | 406.7 | 4 | 0.41 |
| Random | PF0.5_rnd_FS | 1000 | 706.7 | 7 | 0.71 |
| Road | PF0.5_road_HNVf | 1000 | 1300 | 13 | 1.3 |
| Hazard | PF0.5_hzd_HNVf | 1000 | 1316.7 | 13 | 1.32 |
| Random | PF0.5_rnd_HNVf | 1000 | 1330 | 13 | 1.33 |
| Random | PF0.5_rnd_HNVf_FS | 1000 | 1493.3 | 15 | 1.49 |
| Hazard | PF0.5_hzd_HNVf_FS | 1000 | 1580 | 16 | 1.58 |
| Road | PF0.5_road_HNVf_FS | 1000 | 1780 | 18 | 1.78 |

The scenarios combining PF with fire-smart forest conversion ('PF_FS') show lower effects on the reduction in wildfire areas when compared to the other combined scenarios (Figure 5). In the BRGX, implementing fire-smart forest conversion coupled with low-level fuel treatment ('0.5' scenarios) does not lead to significant reductions in the wildfire areas. For example, if we maintain this forest conversion approach while increasing the PF effort ('5' scenarios), they achieve the same performance as scenarios using PF alone ('PF5').

Regarding the results of the scenarios combining PF with HNVf, it is evident that the '5' scenarios continue to yield the best performance (Figure 5). In summary, when the focus is on random landscape management, reductions of approximately 28% are expected. In a scenario aimed at fire hazard mitigation under the same fire-effort conditions ('PF5_hzd_HNVf'), reductions of 31% are anticipated (Figure 5 and Table 4). These scenarios of HNVf combined with PF (even only burning 1000 hectares per year) could achieve reductions that surpass scenarios that exclusively rely on PF as a management strategy, which would burn up to 10,000 hectares per year ('5' scenarios, see Figure 5).

Lastly, the combined scenarios of PF with HNVf and FS would significantly reduce the potential area to be burned under the BAU scenario (see 'PF5_HNVf_FS' scenario in Figure 5). In the study area, in scenarios with a lower FP effort ('0.5' scenarios), annual reductions of around 1500 hectares are expected (in the 'random' and 'hazard' scenarios, i.e., reductions of 15 and 16% over 30 years, respectively, see Table 4), and 1800 hectares per year are expected in the 'road' scenario (with a reduction of 18% over 30 years of simulations, see Figure 5 and Table 4). Once again, the allocation of PF ('5' scenarios) following a fire hazard-oriented management approach showed the best performance, followed by the landscape-oriented approach (see scenarios 'PF5_hzd_HNVf_FS' and 'PF5_rnd_HNVf_FS', respectively, Figure 5). In summary, burning 10,000 hectares/year

with PF, converting 250 hectares/year into farmland, 800 hectares/year into pastures, and promoting the expansion of fire-smart species may reduce the impact of fires in RBGX by 26–32%, regardless of the PF allocation strategy, i.e., annual reductions of 2600–3200 hectares (Figure 5 and Table 4).

Regarding leverage in the combined scenarios, our results point to higher values in scenarios that combine all landscape transformation options ('PF_HNVf_FS'), followed by the prescribed fire scenarios with the promotion of agricultural areas and lower values in scenarios that combine fire with a fire-smart conversion (Table 4).

## 4. Discussion

This study provides compelling evidence for the effectiveness of PF and its integration into landscape management for mitigating wildfire impacts in a region characterized by high fire susceptibility.

### 4.1. Effectiveness of Prescribed Fire Planning at Reducing Future Wildfires

Overall, our results underscore that the low PF effort currently practiced (0.5% annual treatments) will not have a significant medium- to long-term influence on reducing wildfire areas [35,40]. On the contrary, an increase in the annual fuel treatment rate of up to 5–10% of the landscape would significantly restrict the wildfire extent [28].

The ideal scenario with a 5% prescribed fire effort would achieve very promising results (i.e., 11–21 times better compared to current practice) over a 30-year period. The promotion of landscape mosaics and fire hazard management were the strategies that exhibited the best performance. Simultaneously, our simulations revealed that taking advantage of the road networks, i.e., containing wildfires at defensible locations (burning 200 m along road accesses), might not be an acceptable solution in wildfire mitigation, as it showed the poorest performance. Further testing would be required to determine if increasing the treatment size from 200 m to 500 m (or another value) or using spatial information of road density at higher resolutions would counter these results. As expected, the least-realistic scenarios, namely burning 20,000 hectares per year, had a greater impact on reducing the wildfire extent, once again with better performance in promoting landscape mosaics and fire hazard management.

In summary, our model simulations suggest that (i) that burning 10,000 and 20,000 ha/year across the landscape without spatial criteria (size and/or location), allowing the creation of open spaces and fuel discontinuity, and (ii) burning areas with medium to high fuel loads, i.e., areas that have not burned over a certain period of time, would mitigate the impact of future wildfires by around 21 to 36% by 2050 (scenarios 5 and 10%, respectively). These results align with other studies [66,67], indicating that fire hazard-oriented planning would increase the leverage from 0.18 to 0.32, particularly when the effort is low (see 0.5% scenarios in Table 3) [68]. In fact, the leverage values approximate to an overall value of 0.28 estimated for mainland Portugal [63].

Our results broadly support the use of fire as a management tool toward making landscapes more resistant to future wildfires [36,37,57,69]. The spatial extent of PF treatments is critical regarding the disruption of the wildfire rate of growth and the consequent decrease in area burned [28].

### 4.2. The Integration of Prescribed Fire Planning into Land-Use Policies

We assessed the potential evolution of future wildfires within the current context of rural abandonment, primarily driven by the loss of farmland and agropastoral activities. This landscape transformation fuels a fire regime that is expected to become increasingly severe in the near future [16].

We aimed to test the performance of prescribed fire combined with different landscape transformation options under the same treatment allocation strategies, with two levels of effort (the currently practiced 0.5% and the more desirable 5%). Regarding prescribed fire allocation strategies, our simulations indicate that taking advantage of the road network

(firefighting-oriented strategy) continues to be the strategy with the lowest performance. However, its performance improves when combined with landscape-scale transformations (3–4 times higher in '0.5' scenarios and double in '5' scenarios when combined with 'HNVf' and 'HNVf + FS'). Our results suggest that a landscape-oriented approach (i.e., random burning) combined with HNVf areas and HNVf areas with FS conversion will have greater overall effects on reducing wildfire areas over a 30-year period than under BAU scenarios. Finally, once again, focusing treatment efforts on areas with a medium–high fire hazard appears to be the option with the greatest impact on future fire mitigation, especially when combined with landscape transformations: HNVf and HNVf with FS. Promoting FS areas alongside PF does not seem to perform as well compared to other scenarios, regardless of the allocation strategy or effort applied.

As mentioned earlier, scenarios involving an annual burning of 10,000 hectares yield the highest effectiveness in reducing wildfire extent, particularly in the 'random' and 'hazard' strategies. However, in terms of their leverage, these scenarios exhibit intermediate performance, ranging from 0.28 to 0.32, which implies that the wildfire extent can be decreased by 1 ha by treating approximately 3 ha. This leverage decreases in scenarios that utilize the road network, regardless of the fire-effort or land-use combination. Leverage begins to increase when PF is used at a rate of 1000 ha/year in combination with 'FS' in the 'hazard' and 'random' strategies, with leverage values of 0.41 and 0.71, respectively. Leverage increases significantly in the 'HNVf' scenarios and nearly doubles in the 'HNVf + FS' scenarios, for the same effort level of 1000 hectares per year, regardless of the treatment allocation strategy. These results indicate a higher leverage in forested areas (scenarios that include fire-smart conversion), and this leverage tends to increase when combined with agriculture, as previously noted by other analyses of fuel dynamics and of fire behavior and fire severity [34,35,65].

Our findings suggest that a holistic approach, where wildfire management comprising PF is integrated into regional land-use policies [32,52,57], would promote landscapes that are more resilient to future wildfires. In brief, this paradigm shift underscores the pressing requirement for a comprehensive landscape-level management alteration, primarily geared toward minimizing the effects of wildfires. To this end, we demonstrate how prescribed fire can facilitate these transformative processes. It is worth highlighting that prescribed fire entails the use of controlled, low-intensity fires for specific objectives, including but not limited to reducing fuel accumulation [70], protecting assets and people, defending forests against wildfires [66], promoting open spaces, and fostering a biodiverse ecosystem. We emphasize that future agroforestry policies (e.g., the EU-CAP Policy) should support agricultural and forestry activities that allow rural communities to live in a sustainably way. These activities would also contribute to creating heterogeneous landscapes, which are more resilient to large wildfires [71].

### 4.3. Fire Policy Insights under Changing Landscape

Our findings agree with other studies that underline that fire exclusion increases the overall susceptibility of the landscape to severe wildfires [5,72]. Numerous studies have shown that fire is a natural process present in various ecosystems, providing benefits to flora and fauna, making their complete exclusion from the system impossible and ecologically detrimental [34,73]. We have shown that fuel reduction can occur under specific strategies with varying levels of fire-use intensity. On one hand, we can use fire alone to manage wildfires, but it would require a substantial increase in the annually treated area. Due to the significant policy-related and practical challenges associated with PF, fire management policies should adopt a pragmatic approach and consider unplanned fires as valuable complements to PF strategies [32,34]. On the other hand, promoting policies that encourage farming and fire-smart forests [12,57], along with fuel treatments, such as PF, can significantly mitigate wildfires. Regarding the allocation of the fire effort, our study reveals that strategies focused on creating fuel break networks (aiming to contain wildfires and enable direct suppression) have the lowest effect, regardless of increased

PF use, when compared to the other strategies studied here. Managing high-hazard fire areas, in combination with other landscape management strategies, appears to be a cost-effective mitigation action. This hazard-oriented strategy, along with agricultural and forest conversion, seeks to simultaneously eliminate areas with high fuel loads, create heterogeneous mosaics in the landscape [74], and facilitate suppression efforts.

In this way, integrative land management strategies that take into account the various social, economic, and ecological dimensions of territories offer effective solutions for highly populated landscapes in a changing future.

*4.4. Future Perspective and Challenges*

Protected areas are established with the primary purpose of safeguarding natural resources, yet the implementation of strict protection measures often overlooks the vital role of fire in shaping these ecosystems and the traditional use of fire by local communities [27,54,56]. To enhance the effectiveness of protected areas, it is imperative to consider fire suppression and the role of fire as a crucial tool [56]. By introducing PF into these landscapes, a connection can be established between traditional land management practices and the long-term environmental sustainability of complex socioecological systems, such as rural mountain areas [75].

Our model allowed us to test the effectiveness of PF in different alternative landscape scenarios. The reduction in burned areas achieved through these combined scenarios highlights the growing importance of managing, shaping, and sustaining heterogeneous landscapes with a relevant proportion of open spaces driven by fire and of traditional agroforestry activities that can withstand future large wildfires. We are aware of the lower acceptance of fire as a management tool compared with other options [76]. This social resistance creates risk aversion and barriers (social) and limitations (legal) [77,78] for the work of organizations that use fire as a management tool in our region. The current consensus among fire researchers and land managers on the need for more balanced fire management has not yet been translated into policies due to this lack of social acceptance [79]. Effective communication and promotion, education, and demonstration projects are needed to raise awareness regarding the ecological role of fire and the benefits of PF [34]. Fire management can be conceived as a way to achieve a certain fire regime that benefits both ecosystems and humans [5,80]. Our findings underline that the most effective approach toward controlling the fire regime is to promote landscape-scale fuel modification, achieved through both a strategic use of fire and land-use conversion. Implementing these mitigating strategies will contribute to prevent future fire regimes from being mostly controlled by climate [4,81].

As highlighted in previous studies, such as [34], the importance of long-term policies aimed at addressing the structural causes of wildfires and integrating wildfire and forest management strategies is paramount [5,82]. Our model simulations point toward an urgent paradigm shift, centered on 'fire-smart' management, which should aim to mitigate fire severity through fuel treatments accompanied by land-use conversions (preferably to fire-resistant or fire-resilient vegetation types). This change will require the valorization of rural spaces, the reintroduction of agropastoral activities, the recognition of traditional knowledge and practices of the local population, such as shepherds and traditional fire users [36,75], the promotion of resources, the support for population settlement to drive this change at local levels, and a shift in how wildfires are managed, fought, and prevented. The first step is to raise awareness among organizations and public decision makers about the growing need to act at the landscape scale to reverse current policies that contribute to the escalating cycle of highly flammable landscapes.

## 5. Conclusions

There is a pressing need for the re-evaluation of fire within the Mediterranean ecosystem dynamics, recognizing it as an intrinsic element. This recognition should facilitate a paradigm shift toward developing adaptive fire management strategies that prioritize the reduction in adverse wildfire impacts, rather than eliminating this disturbance from the

system. The promotion of strategies that encompass a combination of diverse fuel management practices (such as grazing or promotion of agriculture) holds significant promise in the integration of social, economic, and ecological dimensions for building resilient landscapes and coexistence with fire. This approach not only addresses immediate fire risks but also contributes to ecosystem health and sustainability.

Our results show that using fire as a landscape management tool, combined with the HNVf, could significantly reduce the extent of future wildfires (with reductions ranging from 13% to 31% over 30 years of simulations). This impact is even more-pronounced when combined with 'fire-smart' forest conversions. In other words, our study demonstrates that in the presence of a diverse landscape consisting of fire-resilient agricultural and forest mosaics, we can mitigate the impact of wildfires through prescribed fire (resulting in reductions of 15% to 32% over 30 years).

Action at the level of the fire hazard-oriented approach proved to be the spatial strategy with the greatest effect. The landscape-oriented approach (burning random areas) was the second-best strategy. The following management recommendations ensue from this study results:

(1) *Prescribed Fire*: Its implementation stands out as an effective strategy for reducing future wildfires in our region. Although this approach demands the annual management of a substantial portion of the landscape (5%/year), its wildfire prevention benefits are substantial.

(2) *Strategic Spatial Allocation*: Areas where fuel has built up (those that did not burn within a specific time frame—hazard areas) should be considered in integrated fire management processes. Continuing to create black corridors with the aim of creating fuel breaks may not be the most effective solution. Landscape-scale management, on the other hand, has shown great promise.

(3) *Fire-Smart Land-Use Policies*: Combining prescribed burning with "fire-smart" land-use policies, such as the promotion of High Nature Value farming and the restoration of native woodlands, can significantly enhance efficiency. This synergistic approach can reduce the area requiring annual treatment.

These recommendations aim to not only mitigate wildfires but also optimize resource allocation and bolster landscape resilience.

**Author Contributions:** Conceptualization, S.P. and A.R.; methodology, S.P., N.A. and A.R.; software, N.A. and S.P.; validation, S.P.; formal analysis, S.P., N.A. and A.R.; investigation, S.P.; writing—original draft preparation, S.P.; writing—review and editing, S.P., N.A., A.R., J.P.H. and P.M.F.; supervision, A.R., J.P.H. and P.M.F. All authors have read and agreed to the published version of the manuscript.

**Funding:** This research was partially funded by Portuguese national funds through FCT—Foundation for Science and Technology, I.P., under the FirESmart project (PCIF/MOG/0083/2017) and by MCIN/AEI through the project GREENRISK (PID2020-119933RB-C22). A.R. and N.A. are funded by the Spanish Ministry of Science and Innovation (IJC2019-041033-I and FCJ2020-046387-I, respectively). S.P. received support from the Portuguese Foundation for Science and Technology (FCT) through the Ph.D. grant 2020.09853.BD. P.M.F. was supported by National Funds from FCT—Portuguese Foundation for Science and Technology, under the project UIDB/04033/2020.

**Institutional Review Board Statement:** Not applicable.

**Informed Consent Statement:** Not applicable.

**Data Availability Statement:** REMAINS package and data available on https://github.com/FirESmart-Project/REMAINS (accessed on 1 November 2023). Requirements: computer with R Studio. More information at https://doi.org/10.1016/j.envsoft.2023.105801.

**Conflicts of Interest:** The authors declare no conflict of interest. The funders had no role in the design of the study; in the collection, analyses, or interpretation of data; in the writing of the manuscript; or in the decision to publish the results.

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
