# Peer review of "Optimizing Wildfire Prevention through the Integration of Prescribed Burning into ‘Fire-Smart’ Land-Use Policies"

_fire, doi:10.3390/fire6120457_

Round 1

Reviewer 1 Report

Comments and Suggestions for Authors

My only concern is that some pictures are really difficult to read. For example, what are the 3 figures in Page 6 standing for? Fig. 4-5, what are the meanings of scatters not linked to lines? 

The paper is a good work.

Comments on the Quality of English Language

good.

Author Response

Thank you for your feedback and for reviewing our manuscript! We appreciate your expression of concerns. Regarding the three images on page 6, they have been included in Table 1 as a supplementary and informative element illustrating the spatial distribution of the three prescribed fire strategies. Our intention with these images is to provide the reader with a visual and spatial understanding of the allocation of potential treatment areas.

Regarding the question about Figures 4 and 5, the scattered points with no connection to lines correspond to random data points with no discernible pattern or trend. Each scenario was replicated multiple times, and these points represent those replications. Once again, our gratitude.

Reviewer 2 Report

Comments and Suggestions for Authors

Dear authors, 

Your manuscript is well-written, with clear results, discussions and conclusions. 

My concern is regarding your article The REMAINS R-package: Paving the way for fire-landscape modeling and management which is similar with this new one. Can you clarify my concern?

Author Response

We appreciate your review and positive feedback. In response to your concerns, we would like to clarify that this manuscript aimed to test well-defined and grounded scenarios focused on the current issue of wildfires and the importance of landscape management. Meanwhile, the article "The REMAINS R-package: Paving the way for fire-landscape modeling and management" had as its primary objective the presentation of the REMAINS model and its simulation capabilities. Once again, our gratitude.

Reviewer 3 Report

Comments and Suggestions for Authors

The authors present an article that contributes to the knowledge of how fuel treatments through prescribed burning can contribute to preventing wildfires. The language used is clear, concise and appropriate for an academic article. The research methodology is presented clearly and in sufficient detail. The authors present research results that can inform land-use and fire management policies. The impact and relevance of the results not only refer to the potential of carrying out prescribed burning and promoting mosaic landscapes, but also show the limited effectiveness that reducing fuel on the sides of the road network as a wildfire management strategy can have.

The tables and figures are correct and legible. Nevertheless, the authors should include the authorship of the photograph that is included in Figure 1.

Author Response

Thank you for your comments and for reviewing our manuscript. We appreciate your positive feedback. Regarding the authorship of the photograph in Figure 1, we have already made the necessary changes in the final version of the manuscript. Once again, our gratitude.

Round 2

Reviewer 2 Report

Comments and Suggestions for Authors

I understand your point of view. 

Author Response

Thank you for your positive feedback.